# Temporal Instability of Motorcycle Crash Fatalities on Local Roadways: A Random Parameters Approach with Heterogeneity in Means and Variances

**DOI:** 10.3390/ijerph20053845

**Published:** 2023-02-21

**Authors:** Thanapong Champahom, Chamroeun Se, Sajjakaj Jomnonkwao, Tassana Boonyoo, Amphaphorn Leelamanothum, Vatanavongs Ratanavaraha

**Affiliations:** 1Department of Management, Faculty of Business Administration, Rajamangala University of Technology Isan, Nakhon Ratchasima 30000, Thailand; 2School of Transportation Engineering, Institute of Engineering, Suranaree University of Technology, Nakhon Ratchasima 30000, Thailand; 3Traffic and Transport Development and Research Center (TDRC), King Mongkut’s University of Technology Thonburi, Bangkok 10140, Thailand

**Keywords:** heterogeneity in means and variances, rider characteristics, local roads in developing countries, temporal stability, motorcyclist fatality

## Abstract

Motorcycle accidents can impede sustainable development due to the high fatality rate associated with motorcycle riders, particularly in developing countries. Although there has been extensive research conducted on motorcycle accidents on highways, there is a limited understanding of the factors contributing to accidents involving the most commonly used motorcycles on local roads. This study aimed to identify the root causes of fatal motorcycle accidents on local roads. The contributing factors consist of four groups: rider characteristics, maneuvers prior to the crash, temporal and environmental characteristics, and road characteristics. The study employed random parameters logit models with unobserved heterogeneity in means and variances while also incorporating the temporal instability principle. The results revealed that the data related to motorcycle accidents on local roads between 2018 and 2020 exhibited temporal variation. Numerous variables were discovered to influence the means and variances of the unobserved factors that were identified as random parameters. Male riders, riders over 50 years old, foreign riders, and accidents that occurred at night with inadequate lighting were identified as the primary factors that increased the risk of fatalities. This paper presents a clear policy recommendation aimed at organizations and identifies the relevant stakeholders, including the Department of Land Transport, traffic police, local government organizations, and academic groups.

## 1. Introduction

Thailand has the highest rate of road accident fatalities in ASEAN and ranks ninth globally, with an estimated death rate of 32.7 per 100,000 people, as per the projections made by the World Health Organization [1]. The United Nations initiated the Sustainable Development Goals campaign, and the Cities Development Group identified a specific goal aimed at improving road safety and effectiveness [2]. Typically, the fatality rate resulting from accidents is the primary metric for assessing road safety. Regarding road accidents, crashes involving motorcycles are the type that most commonly leads to fatalities [3]. The prevalent use of motorcycles in low- to moderate-income countries resulted in this problem, which is likely to worsen with an increase in motorcycle registrations [4,5]. Numerous studies identified various causes of these crashes, such as speeding and sudden lane changes [6]. Motorcyclists are considered vulnerable road users (VRUs) due to a lack of protective gear and their limited ability to protect themselves. Therefore, the occurrence of a crash, such as a collision with a fixed object at high speed or with another vehicle, significantly increases the risk of fatalities [7].

There are three main road types in Thailand. First, arterial and highway roads refer to roads that connect provinces or districts. These usually consist of a large number of lanes focused on mobility, and they have heavy traffic volumes. In Thailand, these roads are maintained by the Department of Highways. Second, collector and distributor roads focus on connecting the highway to the district or to connecting districts with 2–4 traffic lanes and a low number of cars [8]. This group of roads is supervised by the Department of Rural Roads. Third, local roads focus on accessibility while entering the village. They may connect to both highways and distributor roads; thus, their traffic volume is low with a small number of traffic lanes, and there are large numbers of trucks in their traffic. Regarding the total length of roads, local roads are the longest road type, comprising approximately 86% of the total road length (Figure 1) [9].

Considering only motorcycle-related crashes classified by road type in Figure 2, collisions on highways have higher fatalities than those on local roads for many reasons, including the numerous vehicle types on highways, both medium and large in size. Motorcycle crashes on highways have a high death rate. Moreover, the speed used on highways is much higher than that on local roads [10]. However, the majority of accidents occur on local roads. The cause for the abundance of accidents on these roads stems from the fact that the majority of motorbikes are utilized in such areas. To begin with, these regions lack a public transportation system, and their centers of activity are relatively remote and serve low-income populations, thereby necessitating the widespread use of motorcycles. Additionally, traffic laws in these regions are not strongly enforced, leading to a high number of novice and youthful drivers who lack knowledge of road regulations. For example, young riders frequently participate in high-speed riding, which can lead to deaths in the surrounding area [11]. Many studies have analyzed motorcycle crashes on highways [12,13]. However, limited research has been conducted on local roads, where crashes are triggered by multi-risk causes due to low traffic levels, such as the use of high speeds by motorcyclists and the fact that these roads are in local areas with minimal law enforcement. Thus, violations frequently happen due to multiple risk factors, including riders exceeding the speed limit, failing to use the signal light before turning, or even being a young rider without a license [14].

Most studies on road accidents utilize crash data and typically employ an analytical method based on the dependent variable’s characteristics, such as the severity of injuries sustained. As a result, a logit model is often utilized as the statistical model [15]. In estimating the parameters, “random parameters” are often allowed to have varying effects across crash populations. For this study, differences in each crash that cannot be observed were taken into account by applying the principle of unobserved heterogeneity in means and variances of the random parameters. Previous research demonstrated that a random parameter logit model with unobserved heterogeneity in means and variances is the most appropriate model to use [16,17,18]. Fu and Sayed [19] analyzed the traffic conflict-based crash estimation, and they found that the model that captured unobserved heterogeneity in means and variances outperformed the standard models in terms of goodness of fit, explanatory power, and crash estimation accuracy and precision. In addition, to enhance the accuracy of the model and better account for complex data, an additional aspect to consider is temporal instability. One study hypothesized that the severity of motorcycle accidents on local roads will fluctuate over time due to several factors, such as improvements in motorcycle brake systems that lead to shorter braking distances and lower injury severity [20]. Furthermore, an effective campaign implemented by local government organizations to reduce accidents, promote the use of helmets, encourage responsible driving practices, and educate younger drivers on safe driving will significantly lower the risk of fatal accidents over time [21,22]. Yu, et al. [23] confirmed that temporal instability was a significant factor for rear-end crashes in work zones between 2010 and 2011 and 2012–2013. Yan, et al. [24] compared the crash severity of male and female riders and also accounted for temporal analysis; they found the effects of different genders were nontransferable and underscored the necessity to accommodate temporal influences.

Table 1 presents research that focuses on the severity of motorcycle accidents, with specific attention given to various types of crashes, including single- and multiple-vehicle crashes [25]. Some studies analyzed accidents in specific areas, such as urban and suburban road areas, arterial roads, and intersection areas [10]. Other research investigated various analytical methods, including both machine learning and discrete models [26]. As seen in Table 1, the majority of studies utilized a random parameter model with heterogeneity in means and variances for analysis. However, no prior research has been conducted on motorcycle accidents that occur on local roads. These roads are statistically less fatal than highways but are the longest compared to other road types, and they are frequently used by motorcyclists. Therefore, research focused on motorcycle accidents on local roads should be conducted promptly.

The objective of this study was to examine the risk factors that have an impact on the fatality rate in motorcycle accidents on local roads. Additionally, because motorcycle crash severity may also exhibit temporal variations, this study additionally considered the concept of temporal instability. By shedding light on the underlying factors contributing to motorcycle accidents, this study can aid in developing effective policies and recommendations to reduce the severity of such accidents on local roads. The relevant agencies responsible for constructing and maintaining the roads, as well as those in charge of road safety, can use the results of this study to implement appropriate measures and recommendations.

The present research article is divided into several sections. The first component is a review of factors that may influence the severity of motorcycle crashes. This is followed by a discussion of the analytical methods and of model development. The subsequent section details the dataset and its preparation for analysis. The next section is called the Temporal Stability Test, which was used to evaluate the model’s suitability for each year and over a three-year period. The results section provides an overview of the preliminary analysis findings. The discussion section provides a comprehensive argument supporting the model’s validity and explaining the various crash factors, such as rider characteristics, maneuvers prior to the crash, and temporal and environmental factors. The final section summarizes the study results and offers policy recommendations based on the model’s findings, along with application guidelines.

## 2. Literature Review

Table 2 summarizes the four groups of factors that are expected to influence the severity of injury levels on local roads, which include rider characteristics, maneuvers prior to the crash, temporal and environmental characteristics, and road characteristics.

### 2.1. Rider Characteristics

According to previous research, the rider is consistently identified as the primary factor contributing to accidents. This is determined by various variables that can be grouped into different dimensions. For instance, differences in collision speeds caused by different genders can lead to varying levels of injury severity [34]. Younger riders, due to their energy and inexperience, are more likely to drive at high speeds, although their physical strength can reduce their level of injury in the event of an accident [37]. The rider’s address, whether local or foreign, is also a factor that can impact accident severity. Foreign riders in Thailand may not be familiar with the roads, particularly around intersections, leading to a higher likelihood of accidents [33]. Despite the low traffic density on local roads, motorcycles are often driven at high speeds, which increases the risk of accidents [51]. Wearing helmets can lower the level of injuries sustained by motorcyclists [29]. Finally, possessing a rider’s license indicates that the motorcyclist has the necessary driving skills and is more likely to follow traffic rules, which in turn can reduce the likelihood of accidents [42].

### 2.2. Maneuvers Prior to the Crash

This group of factors is considered to be the underlying causes of crashes and encompasses several factors, such as exceeding the speed limit. This variable is present in crashes that occur at high speeds where the vehicle is thereby unable to brake in time, especially when colliding with fixed objects such as light poles, kilometer markers, and semis. Such crashes often result in more severe collisions [7]. Another variable is driving in the wrong direction, which is rare. However, when such a crash does occur, it almost always involves a head-on collision with high impact [44]. Sudden cutting across lanes is one of the most common causes of crashes, particularly near local roads with numerous entrances and exits. In addition, intoxicated riders are a significant variable, as they are more likely to make decisions more slowly than non-intoxicated individuals, potentially increasing the severity of injuries sustained [41].

### 2.3. Temporal and Environmental Characteristics

This group of factors pertains to the temporal and environmental factors that may affect traffic volume and road user characteristics. The duration of a festival that coincides with a long holiday has an influence on crash characteristics. As a result, there are many intoxicated motorcyclists on local roads who drive at high speeds with delayed reaction times, potentially increasing the severity of crashes [27]. During rush hours, this factor may not be as noticeable on local roads as it is on city roads because locals do not typically work for extended periods. However, in the morning, there is an increase in traffic that results in reduced driving speeds, and there are often many students going to school. Therefore, a group of young riders tends to be more prevalent on local roads during this time [46].

The rider’s ability to perceive and make decisions to reduce speed and prevent crashes is impacted by their visibility. This group of factors includes the time of day, such as daytime or nighttime. During the day, visibility is generally good, but the severity of daytime crashes may be higher than those at night due to higher speeds used by motorcyclists [30]. Another significant variable is crashes that occur at night with and without lights, which has been highlighted in numerous studies. Crashes with no lights can increase the severity of injuries because vehicles or objects in front may not be visible from a distance, leading to delayed braking and higher impact force during a collision [46]. Rainy weather is also a factor that affects crash severity, as it can impact visibility and vehicle handling due to slippery roads. This may cause motorcyclists to decrease their speed, ultimately reducing the severity of crashes [32].

### 2.4. Road Characteristics

The readiness and physical characteristics of the road surface can have a significant impact on driving speed and, ultimately, the severity of crashes. This category of factors includes various examples such as road maintenance, where closed traffic lanes can cause sudden braking and result in a potential collision [16]. Another factor to consider is a rough road surface with potholes, which can lead to serious crashes if a motorcyclist is traveling at high speeds [43]. Additionally, curves in the road can increase the likelihood of motorcycle accidents, particularly if a person is unfamiliar with the route or enters a curve at high speed, causing the vehicle to veer off and result in severe injuries [50]. The road surface’s condition, which can be either wet or dry, is another variable in this group. Although wet roads may occur during or after rainfall, the severity of crashes on dry roads is typically higher because motorcycle riders tend to be more confident and travel at higher speeds on these surfaces [47].

## 3. Methods

The main objective of this study was to establish the relationships between the independent variables and the dependent variables that contribute to the death of a motorcycle rider. Injury severity was categorized into two groups: fatal and non-fatal. To analyze the data, the study employed a statistical modeling approach using the concept of random parameters that account for heterogeneity in both means and variances. The parameter estimation first introduces a function that assesses the severity of the crash
(1)$$S_{jm}=\beta_{j}X_{jm}+\varepsilon_{jm},$$
where $S_{jm}$ is the function of crash *m* sustaining injury *j*, $X_{jm}$ is a vector of the explanatory variable (such as rider characteristics, cause of crashes, temporal characteristics, and environmental characteristics), and $\varepsilon_{jm}$ refers to the error term. To account for unobserved heterogeneity, the parameters were allowed to vary across observations. The probabilities function of the random parameters model can be defined as follows:(2)$$P_{m}\left(j\right)=\int \frac{EXP\left(\beta_{j}\;X_{jm}\right)}{\sum_{\forall j}EXP\left(\beta_{j}\;X_{jm}\right)}f\left(\beta |\rho \right)d\beta ,$$
where $P_{m}\left(j\right)$ is the probability of crash *m* sustaining an injury severity *j*,$\;f\left(\beta |\rho \right)$ is the density function of β, and ρ refers to the vector of the parameters (mean and variance). To further allow the explanatory variable to affect the distribution of the random parameters (mean and variance), $\beta_{jm}$ is treated as a vector of an estimable parameter that is allowed to vary across crashes, expressed as
(3)$$\beta_{jm}=\beta_{j}+\Theta_{jm}Z_{jm}+\sigma_{jm}EXP\left(\omega_{jm}W_{jm}\right)\nu_{jm},$$
where $\beta_{j}$ represents the mean of parameter estimation, $Z_{jm}$ is a vector of the explanatory variable that captures the heterogeneity in means, with $\Theta_{jm}$ as a corresponding vector, $W_{jm}$ denotes a vector of the explanatory variable that captures heterogeneity in variance (i.e., standard deviation, $\sigma_{jm}$), with $\omega_{jm}\;$ as a corresponding vector, and $\nu_{jm}$ denotes a disturbance term.

The McFadden $\rho^{2}$ statistic was utilized to determine the ideal model selection based on its proximity to empirical data [52] as follows:(4)$$\rho^{2}=1-\frac{LL\left(\beta \right)}{LL\left(0\right)}$$
where $LL\left(\beta \right)$ is the log-likelihood of the model with all parameter estimates and $LL\left(0\right)$ is the log-likelihood of the model with constants only [53]. Although there is no definite definition of the optimal value, road crash models may be deemed satisfactory at values greater than 1 [54]. With random parameter analysis, the McFadden $\rho^{2}$ values can significantly increase; however, this may also result in a higher number of estimated parameters. As a result, the adjusted McFadden $\rho^{2}$ metric should be analyzed:(5)$$adjusted\;\rho^{2}=1-\frac{LL\left(\beta \right)-k}{LL\left(0\right)}$$
where *k* is the number of estimated parameters [55]. Furthermore, the average marginal effect (ME) across all crash observations was calculated to determine how a one-unit increase in an explanatory variable affects the probabilities of different severity outcomes. This was done to evaluate the impact of individual parameter estimates on the probability of injury severity among riders as the parameters change over time [3].

To develop a random parameter logit model that takes into account unobserved heterogeneity in means and variance, the researchers utilized Nlogit version 6. The Halton sequence of draws was selected for the model’s development, as it provides speed gains without adversely affecting simulation performance [56]. In this study, the model was set to 500 Halton [16].

## 4. Empirical Setting

This study analyzed data from accidents that took place on Thai local roads maintained by the Royal Irrigation Department, local government organizations, municipalities, or Bangkok Metropolitan from 2018 to 2020. Crashes that occurred on highways and rural roads were not included in the dataset. Prior to the analysis, a binary code was employed to prepare the data in which 1 indicates a certain value, and 0 indicates all other values. The definitions of the variables can be found in Table 3. The presence of multicollinearity in the data was assessed by examining the Pearson correlation coefficients of the independent variables, as depicted in Appendix A. It was determined that all pairs have a correlation value of less than 0.7; thus, it indicates the absence of multicollinearity [57].

The outcome variable, which pertains to the number of fatalities resulting from motorcycle crashes, was scaled down to a single rider, where a value of 1 signifies death and 0 indicates other injury levels, including severe or minor injuries [58]. The independent variables were assigned values of either 1 or 0 to make them easier to interpret, such as assigning a value of 1 to male motorcyclists and 0 to female motorcyclists, where 1 represents “yes” and 0 represents “no.” In the case of the young variable, a value of 1 indicates that the rider is under 30 years of age, etc.

Table 3 provides a summary of the crash data for each year, including descriptive statistics. The number of fatal crashes was the highest in 2019, but it decreased in 2020 due to the COVID-19 pandemic (as seen in Figure 2). The most non-fatal crashes occurred in 2018. Various factors are presented in terms of their mean and standard deviation. Of note is the helmet variable, which had a relatively low value of 0.517 and 0.519 in 2018 and 2019, respectively. In 2020, the proportion of riders wearing a helmet further declined to 0.375. Furthermore, the incidence of accidents involving drunk riders increased from 2019 to 2020.

## 5. Temporal Stability Test

The Temporal Stability Test gauged the dissimilarity between crash datasets for each year using the likelihood ratio test principle, which was employed to compare two distinct years as follows [59]:(6)$$\chi^{2}=-2\left[LL\left(\beta_{m_{2}m_{1}}\right)-LL\left(\beta_{m_{1}}\right)\right]$$
where $LL\left(\beta_{m_{2}m_{1}}\right)$ is the log-likelihood at the convergence of the model containing significant (converged) parameters from m_2_ and using data subset m_1_ at the same time. The tests were also reversed for each pair. This process was repeated six times until six pairs were completed. The null hypothesis for the likelihood ratio tests was that the impact of various factors on injury levels over two years was not different (or was the same). The level of confidence was determined using the $\chi^{2}$ statistics, with degrees of freedom equivalent to the number of estimated parameters [59].

Table 4 displays the outcomes of the likelihood ratio test for temporal instability. When converged variables from 2018 were chosen and the data for 2019 and 2020 were analyzed, the null hypotheses were rejected at confidence levels of 95.16% and 98.71%, respectively. Additionally, when significant parameters from the 2019 model were utilized, and when the data for 2018 and 2020 were examined, the null hypotheses were rejected at confidence levels of 95.76% and 99.99%, respectively. Similarly, for the set of significant parameters from the 2020 model analyzed with the 2018 and 2019 data, the null hypotheses were rejected at confidence levels of 98.75% and 99.99%, respectively. These results indicate that the crash data from 2018 to 2020 were not consistent across years, implying a significant degree of “temporal instability”. Therefore, the optimal statistical model for this dataset should account for this segregation.

Another assessment involved determining whether a model was more fitting when partitioned into yearly intervals as opposed to model development utilizing the entire dataset [60]. The likelihood ratio test was also utilized in the analysis, and it was computed as follows:(7)$$\chi^{2}=-2\left[LL\left(\beta_{2018-2020}\right)-LL\left(\beta_{2018}\right)-LL\left(\beta_{2019}\right)-LL\left(\beta_{2020}\right)\right]$$
where $LL\left(\beta_{2018-2020}\right)$ is the log-likelihood at the convergence of the model using the dataset for the years 2018–2020. The number of degrees of freedom (df) is equivalent to the number of parameters identified as statistically significant in the model that uses all of the data from 2018–2020, excluding the number of statistically significant parameters identified in each year. The results of the $\chi^{2}\;$ statistics are displayed in Table 5. The $\chi^{2}\;$ value was 776.542 with a total df of 62, indicating that the null hypothesis should be rejected at a confidence level of 99%. Therefore, years in the dataset should be separately analyzed.

## 6. Results

After evaluating the goodness of fit for the three models, as displayed in Table 5, it was determined that the McFadden *ρ^2^* and Adjusted McFadden *ρ^2^* of the random parameter logit model were more favorable than those of the traditional logit model. The random parameter model, which incorporated unobserved heterogeneity in means and variance, showed McFadden *ρ^2^* values ranging from 0.133 to 0.141. Among the three models, the 2018 model exhibited the highest value, followed by 2020 and 2019. The adjusted McFadden *ρ^2^* values ranged from 0.125 to 0.137, which falls within an acceptable range for interpreting the results and making policy recommendations for road safety [61].

### 6.1. Model Results of MC Crashes in 2018 

In general, a number of parameter estimates were found to be significant. The results of the 2018 analysis are presented in Table 6, showing that the constant value had negative significance, whereas all of the explanatory variables had significant effects. Within the group of rider variables, riders under the age of 30 had a lower likelihood of death, whereas older riders had a higher likelihood of death. The following variable, foreign riders, was found to be associated with a higher risk of death. Similarly, crashes involving helmeted motorcyclists were found to increase the likelihood of more severe outcomes, indicating that helmets may not have met the standard in 2018. However, data from crashes in 2020 revealed that wearing helmets reduced the likelihood of death. The variable of riders living in the same district was associated with a lower chance of death despite being a random parameter. However, the standard deviation of the “local address” variable (the normal distribution) was very small, indicating that almost all collisions with people in the area had a low probability of death. The gender variable, which was insignificant, was found to be a random parameter. No random parameter was found for maneuvers prior to the crash or for hypothetical causes. Exceeding speed limits was associated with a higher likelihood of death, whereas traffic violations, wrong-direction driving, and falling asleep also increased the chance of fatality. Sudden cutting, on the other hand, was found to be a factor associated with a lower risk of death. In terms of temporal and environmental characteristics, the morning peak hour was found to have a higher risk of fatalities, whereas the afternoon peak hour had the potential to decrease fatalities. The festival duration variable was found to be insignificant, but its standard deviation was significant. Crashes during the day were less likely to result in death. Collisions in fog, rain, or smoke reduced the likelihood of death, as did crashes on rough or wet road surfaces. However, crashes on curves were found to be more likely to result in death.

To examine the heterogeneity in the mean of the random parameters, it was found that licensed riders decreased the mean of crashes during festivals, resulting in a lower chance of fatal crashes. On the other hand, drunk riders and dry road surfaces decreased the mean of the gender variable, whereas licensed riders increased it. The mean of the local address variable was increased by the dry road surface variable, indicating a higher likelihood of fatal crashes. In addition, the afternoon peak hour was found to increase the variance of the random parameter of the local address variable.

### 6.2. Model Results of MC Crashes in 2019 

Table 7 presents the 2019 model’s results, which indicate that male riders had a higher likelihood of a fatal injury compared to female riders. In contrast, riders under the age of 30 and those residing in the local area had a lower risk of death. The variable for foreign riders, modeled as a random parameter, showed that 71.4% of the crashes involving foreign riders had a lower probability of death, whereas only 28.6% had a higher probability of more severe injuries (as shown in Figure 3a). The analysis of maneuvers prior to the crash identified two significant variables: mobile phone use while driving and sudden cutting. However, both of these factors had low chances of causing death. In terms of temporal and environmental variables, crashes occurring during the festival period had a relatively low chance of death. When it comes to the timing of crashes, the morning peak had a higher chance of fatalities, whereas the afternoon peak had a lower chance of fatalities. Crashes occurring at night, both with and without lights, were more likely to cause fatal injuries. As for road variables, crashes on rough roads were less likely to cause fatal injuries. Finally, crashes on dry roads were found to have a lower risk of fatalities, but this was identified as a random parameter.

Several factors were found to be affected by other factors in terms of the effect of heterogeneity on the means of the random parameters. Wet road crashes reduced the mean of the festival duration variable (making fatal injury less likely), whereas the helmet variable and daytime collisions increased its mean (making fatal injury more likely). The mean of the variable for crashes on dry surfaces was affected by several factors, including drunk rider, helmet, license, daytime, and age. The drunk rider variable reduced the mean of foreigner involvement in crashes. In terms of heterogeneity in the variance of the random parameters, the variable for riders under 30 reduced the variance of crashes on dry roads. 

### 6.3. Model Results of MC Crashes in 2020

Table 8 displays the results of the 2020 model. The variable for male riders was determined to be a random parameter. Young riders were found to have a lower chance of death, whereas riders over 50 years old had a higher risk of death in the crashes. The variable for the rider having a local address indicated a low probability of death, whereas foreign riders had a greater risk of being killed in the crashes. Wearing a helmet was a significant factor in reducing the likelihood of death, although it was also found to be a random parameter. According to the distribution shown in Figure 3b, only 54% of those wearing helmets had a decreased risk of death, whereas the remaining 46% had an increased risk of death. In terms of the maneuvers made before the crash, most of them were found to have a higher likelihood of causing death, such as running red lights (violation of traffic signal), speeding, and making illegal overtaking maneuvers. On the other hand, sudden cutting and crashes involving intoxicated riders had low chances of causing death. Variables related to drunk riders also produced significant random parameters. However, the standard deviation value for this parameter was relatively small, indicating that the increased risk of death for these factors was only around 1.1%. Crash occurrence during holiday periods was identified as a random parameter. Crashes at night with lights were found to have a higher risk of fatalities. With respect to the features of road infrastructure, roads under maintenance, rough roads, and dry roads, they were identified as factors that increased the risk of fatal crashes.

In terms of heterogeneity in the means of the random parameters, the mean of the festival duration variable was reduced by the indicator for licensed riders, whereas it was increased by the smoke indicator. The mean of the drunk rider random parameter was increased by the daytime and wet road variables, whereas the indicator for crashes that occurred in the rain decreased the mean of drunk riders. The indicators for mobile phone use and rain decreased the mean of the helmet random parameter, whereas those of the curve and license parameters increased its mean. The license parameter also increased the mean value for the gender random parameter. The variance of the helmet variable was affected by the local address and darkness parameters (which increased the variance) as well as the afternoon peak hour (which decreased the variance).

## 7. Discussion

In this section, the parameter estimation results are presented, highlighting the parameters’ effects’ instability over time. The effects of factors are compared using the ME and presented in Table 9 to discuss the results across 2018–2020.

### 7.1. Rider Characteristics

Throughout the three studied years, the data consistently indicated that male riders were more likely to suffer fatalities in the event of a crash. This is likely due to the fact that men tend to drive more aggressively than women, particularly on local roads where policing and law enforcement are minimal. This finding is consistent with previous research by Abrari Vajari, et al. [32]. However, in analyzing the 2018 and 2020 models, it was found that gender was a significant random parameter, indicating that female riders had a lower likelihood of fatality compared to male riders in these models. Over time, there has been a steady increase in the severity of crashes on local roads, as indicated by the ME values which rose from 0.0809 in 2018 to 0.0851 in 2019 and to 0.1143 in 2020. This could be attributed to increasingly aggressive driving behaviors among male riders, such as speeding, sudden lane changes, driving in the wrong direction, etc. [62].

The age factor of riders is comprises two variables: those under 30 years old and those over 50 years old. Young riders had a consistently low chance of fatality each year, whereas older riders had a high chance of death. Despite the fact that young riders tend to drive more aggressively, older individuals are more likely to sustain injuries during crashes, and these injuries can be more difficult to treat [36]. Neither of these factors were found to be random parameters, indicating that the direction of severity was consistent in most cases. With regard to changes over time, young riders experienced a slight decrease in fatal injury probability from 2018 to 2020 (ME = −0.0291, −0.0293, and −0.0517, respectively), whereas older riders had a slightly increased risk of death from 2018 to 2020.

In terms of rider experience and familiarity with the area, the results showed that riders who were from the same district as the crash location had a lower chance of being killed in that crash compared to those from outside the area. This could be due to their familiarity with the roads and traffic, which could help in making quick decisions, thereby reducing the risk of injury. Across time periods, the values were close to each other (ME for 2018–2020 was −0.0116, −0.0139, and −0.0070, respectively for each year). Foreign riders had a high probability of dying in crashes, which may be attributed to their limited knowledge of traffic rules and signs as well as their unfamiliarity with common road behaviors. Additionally, the collisions that occur as a result of sudden cutting in front of foreign riders are more likely to be fatal [40]. In terms of the time period, the 2019 model showed the foreign rider variable was insignificant, whereas in 2020, the effect was significant (ME = 0.0685) and had a larger impact than in 2018 (ME = 0.0479). This variable was not identified as a random parameter; thus, it resulted in a considerably high chance of fatality (as a fixed parameter).

The law-related variables include wearing a helmet and possessing a rider’s license. The analysis results revealed that having a rider’s license was insignificant, indicating that it does not ensure a decrease or increase in the chance of death from a crash, perhaps due to the ease of obtaining a motorcycle rider’s license in Thailand. However, it does help in reducing the frequency of crashes. On the other hand, for the helmet variable, the 2018 and 2020 results were conflicting, and both produced significant random parameters. In 2018, the majority of people wearing helmets had a higher risk of death (ME = 0.0154). The probable reason for this is that helmet wearers tended to be more confident or drive at higher speeds. Additionally, as there are various types of helmets available, local road riders do not prioritize standard and costly helmets, and instead, they purchase non-standard helmets. Therefore, in the event of a crash, the risk of death would be still high. However, the 2020 data demonstrated that the use of helmets reduced the likelihood of death (ME = −0.0756), which is understandable considering the improvement in the quality of riders’ helmets and the intense efforts of helmet-related agencies to spread awareness to the public [63].

### 7.2. Maneuvers Prior to the Crash

The crash characteristics refer to the factors that contribute to the occurrence of the crash. Exceeding the speed limit was found to have a high risk of resulting in fatalities, which is consistent with the findings of Se, et al. [12]. This is due to the fact that speed is the primary cause of crashes for all types of vehicles, particularly on clear roads such as local streets. Regarding temporal stability, the ME value was high in 2018 (ME = 0.1774) and even higher in 2020 (ME = 0.2509). As it was found to be a fixed parameter, it indicates that the speeding violations on local roads continue to occur. Similarly, Alnawmasi, et al. [3] also highlighted that exceeding the speed limit increased the likelihood of fatalities for motorcyclists navigating a curved road.

Next, the variables for crashes that were caused by traffic signal violations include disregarding no-overtaking signs or lines, not yielding at intersections, and running red lights. These variables were found to have a significant positive correlation with the risk of fatal crashes. This may be due to these violations causing collisions with high impacts, such as those occurring at intersections involving different types of vehicles. Regarding wrong-direction driving, there was a higher risk of associated fatalities, which contradicts the findings of Chung, et al. [44], who did not find that it had a significant impact on the severity of motorcycle crashes. However, this is consistent with the research conducted by Gouribhatla and Pulugurtha [64], explaining that it could lead to head-on collisions involving a motorcycle and a larger vehicle that generate powerful forces and can result in a higher risk of riders being killed in those crashes. This variable was not a random factor and was only significant in 2018. Furthermore, illegal overtaking maneuvers were only significant in 2020 and were associated with a higher risk of fatalities. This aligns with the study by Chung, et al. [44], which stated that overtaking in prohibited areas and colliding head-on with an oncoming vehicle increased the chance of riders being fatally injured.

Crashes resulting from mobile phone use had a lower probability of fatalities and were only significant in 2019. The probable explanation for this is that when motorcyclists use their mobile phones while driving, they realize that they cannot effectively control their vehicle, which typically leads them to drive at slower speeds. As a result, the occurrence of crashes may be due to sudden braking or falling over, and the level of severity is typically minor. This is consistent with the findings of Islam [16], who found that when riders were not distracted prior to a collision, their chances of succumbing to fatal injuries from a motorcycle crash increased compared to distracted riders. However, a study by Chung, et al. [44] reported that distractions did not have a significant impact on the injury severity of motorcyclists.

The likelihood of fatalities resulting from falling asleep while driving was only significant in 2018, indicating that motorcyclists rarely fell asleep while riding in the following years. Sudden cutting is another contributing factor to the high frequency of motorcycle crashes due to the motorcycle’s high mobility and the carelessness of riders on local roads. However, this study determined that this factor was unlikely to cause fatalities (ME of 2018–2020 = −0.0227, −0.0817, and −0.0727, respectively for each year). This can be attributed to the fact that most motorcycle riders would likely notice sudden cutting and apply their brakes in time, resulting in less severe crashes. Interestingly, intoxicated riders were discovered to have a reduced likelihood of fatalities in the event of a crash. This outcome contradicts findings from several studies [44]. A possible reason may be that such crashes could include cases where the rider loses control of the motorcycle and falls by themself. However, this variable was a random parameter, suggesting that some of these crashes are less likely to be fatal, but in some instances, they may be more likely to increase the risk of fatal injury. Similarly, Se, et al. [12] discovered that drunk riders significantly decrease fatalities but increase the occurrence of serious injuries in crashes.

### 7.3. Temporal and Environmental Characteristics

The overall variables in this group were found to go in the same direction over the three years studied. Motorcycle crashes were less likely to result in death during a festival period, a variable with a relatively negative ME (ME of 2018–2020 = −0.1226, −0.1368, and −0.1937, respectively for each year). This may be due to the relatively high traffic volume and strict traffic discipline during the festivals, making riders on local roads aware of their speed [1]. However, in 2018–2019, this variable was a random parameter, indicating that certain groups of people who had crashes during the festival duration had a high risk of death due to the increase in intoxicated people compared to non-holiday periods. Therefore, they had slower decision-making processes, which subsequently increased the chance of death in a collision. However, it was not discovered as a random parameter in the 2020 model, which may be attributed to travel restrictions due to the spread of the virus (COVID-19).

The subsequent variable was the timing of crashes. The morning rush hours in 2018 and 2019 were significantly correlated with a high likelihood of fatalities, indicating that crashes occurring between 7 and 8 am had a higher probability of being fatal. This could be due to people being in a hurry to get to work or school and having reduced awareness. Furthermore, accidents involving vulnerable road users (VRUs), such as motorcyclists, would almost certainly result in fatalities during this time of day [65]. However, in 2020, this variable was insignificant, which could be also caused by the travel restrictions. Contrarily, a downward trend in the fatal injury probability of accidents was observed during the afternoon rush hours. This can be attributed to the fact that, although the afternoon rush hour is characterized by heavy traffic, it is not considered a rush hour like the morning period; thus, the chances of high-speed collisions and associated fatalities were reduced. Furthermore, the marginal effect (ME) values slightly decreased in 2019 and became increasingly negative in 2020 (2018–2020 ME = −0.0213, −0.0103, and −0.0513, respectively for each year), indicating a significant decrease in the risk of death in the latest period. In 2020, this decreasing trend can also be attributed to reduced travel during the afternoon rush hours due to the ongoing pandemic, which led to the highest value of ME for that year, indicating a decrease in the probability of fatal collisions. This finding is in line with the conclusions drawn by Chen and Fan [66], who argued that the afternoon peak should be expected to result in less severe injuries compared to the morning peak.

Regarding the time of day, the analysis results indicated that daytime crashes had a lower likelihood of fatalities and were only significant in 2018. This finding is consistent with the study conducted by Li, et al. [30], which showed that daytime crashes reduced the severity of the riders’ injuries compared to nighttime crashes. On the other hand, collisions that occurred at night and with poor lighting in 2019 and 2020 increased the likelihood of fatalities (ME = 0.0853 and 0.0528, respectively). This may be due to the fact that crashes that occur at night when rescuers or other drivers are limited could see delays in notifying rescue efforts, potentially causing help to arrive too late to prevent fatalities [32]. In situations where there was no light, there was a high and significant likelihood of fatalities for all three years (ME of 2018–2020 = 0.0017, 0.0914, and 0.0324, respectively for each year). This could be attributed to the limited visibility making it difficult to see objects, resulting in crashes at high speeds and an increased chance of death. This is consistent with the research conducted by Jama, et al. [67], which demonstrated that crashes involving motorcycles at roadside safety barriers without lights had an increased likelihood of fatalities. In terms of the impact of fog or smoke on visibility, the findings revealed that it resulted in a reduced likelihood of fatalities. This may be due to poor visibility, which likely caused riders to slow down [68]. With regards to weather-related factors, crashes that occurred while it was raining had a low chance of resulting in fatalities, but the ME was relatively small (ME = −0.0005). This was because riders on rainy days were compelled to drive at slower speeds [17].

### 7.4. Road Characteristics

In terms of variables associated with a crash scene on a road undergoing construction or maintenance, it was discovered that the chance of fatalities was significantly high in 2020 with an extremely high ME value (ME = 0.3518) second only to illegal overtaking maneuvers. This suggests that roads under maintenance pose significant danger. The probable explanation is that local roads are not regularly maintained, causing riders to be unfamiliar with the road and receive insufficient warning before reaching the construction site [17]. With regard to the variable of rough road surfaces in 2018 and 2019, the marginal effects were −0.0605 and −0.0143, respectively, indicating a low probability of fatalities. However, in 2020, a significantly positive marginal effect was observed, signifying a higher likelihood of fatal accidents. This may be due to limited maintenance budgets allocated to local roads, resulting in roughness that is not promptly repaired and becomes increasingly hazardous over time. Additionally, accidents on rough roads tend to occur during nighttime hours when lighting is inadequate (as demonstrated by the correlation coefficient in Appendix A), making them more difficult to anticipate and avoid. Consequently, the probability of accidents is higher. However, without reporting temporal instability, Islam [16] similarly posited that smoother road surfaces are safer for motorcyclists.

Regarding curved road sections, they were associated with a greater risk of fatalities, but it was only significant for 2018 and 2019, suggesting that road safety campaigns and inspections were minimal during those years. Many curved roads remain unrepaired, but in 2020, there may be several agencies in charge of handling crashes by checking and improving hazardous areas. Thus, this variable became insignificant in 2020. In terms of the dry and wet road surface variables, the analysis results revealed that dry road surfaces have a high likelihood of causing fatalities, particularly in 2020, potentially because riders in that year were likely to ride at high speeds. This is consistent with the analysis results of the “exceeding the speed limit” variable, which found that if a crash occurred due to this reason, the crashes would be more severe in the 2020 model. These results also align with the findings of various studies, including that of Se, et al. [12], who found that dry surfaces caused riders to drive faster and increased the risk of death in a collision.

## 8. Conclusions

Road crash research has predominantly concentrated on roads with high mobility, such as highways and arterial roads, due to their classifications of accessibility and mobility. However, local roads, which emphasize accessibility, also contribute significantly to fatalities resulting from road crashes. This study aims to examine the factors that influence fatal motorcycle crashes on local roads to improve our understanding of road crashes. The outcomes of this study can be utilized to develop policy recommendations by relevant agencies such as local government organizations, the road safety center, and the Ministry of Public Health to improve motorcycle safety.

This study employed four categories of variables, namely rider characteristics, maneuvers prior to the crash, temporal and environmental characteristics, and road characteristics. To conduct the statistical analysis, a model that took into account the heterogeneity in means and variances of random parameters along with the principle of temporal instability was utilized in the study.

The summaries of the key findings are as follows. The study found that motorcycle-involved crash data on local roads from 2018 to 2020 varied by year. Many variables in each year were found to be statistically significant random parameters, and their means and variances were affected by other considered risk factors. In terms of rider characteristics, male riders had a tendency to increase their risk of being fatally injured over time compared to the female rider counterparts. Riders over 50 and foreign riders were significant variables in 2018 and 2020 that tended to also increase the chance of fatalities. Numerous cause-related factors were found to be positively associated with fatal crashes, including exceeding the speed limit, disobeying traffic signs, illegal overtaking maneuvers, and driving in the wrong direction. These factors may constitute rather worrisome variables, as they were significant in all three-year models with a higher chance of death. On the contrary, some factors from this variable group were associated with a decrease in injury severity levels, including crashes due to sudden cutting in front and mobile phone use while driving. Regarding the time of the crashes, the morning peak hours produced a higher likelihood of fatal crashes, whereas crashes during the afternoon peak hours and during festival occasions were less likely to cause fatal injuries. Regarding environmental factors, most models discovered that low-light conditions or poor visibility would increase the likelihood of a fatal crash (for example, a crash at night with light and one without light). Finally, regarding roadway attributes, curved roads and areas under maintenance or construction were also found to have a higher risk of causing fatal crashes.

## 9. Implementations

To develop policy recommendations, factors that had significant effects or had high and sustained marginal effects over several years were considered. The variable of rider gender showed that male riders were more prone to aggressive behavior than female riders. An actionable policy recommendation could be to present statistical data on gender and the likelihood of death to increase rider awareness of the danger, and to promote awareness of the probability of death while driving on local roads by relevant agencies such as the Department of Land Transport, which is responsible for driver’s license training. For elderly riders, public relations posters or notices should be created to help them understand that their physical attributes are not as robust as those of teenagers, which increases their risk of death in the event of a crash. This could be promoted by relevant agencies such as the Thai Health Promotion Foundation (Thai Health) through media such as television. To educate foreign riders, it is essential to provide knowledge before allowing them to rent a vehicle, such as information on Thai people’s driving behavior, the meaning of warning signs, and the basic traffic rules, such as speed limits. Additionally, related agencies, including the Ministry of Tourism and Sports and the Traffic Police, could take necessary action.

During the morning rush hour, there was a heightened risk of fatalities. It is imperative that a public awareness sign be displayed to emphasize the need for extra caution between 7:00 a.m. and 8:00 a.m. Raising riders’ awareness of the dangers of excessive speed during the morning rush hour is crucial and can be promoted through various sources such as government centers, schools, universities, and other places of activity. The high speeds of motorcycles during the morning rush hour pose a significant risk and should be addressed in training for students and parents. The relevant government agency responsible for road management should also consider implementing measures to control speed during the morning rush hour. These countermeasures should specifically target rushing riders, such as by posting speed limit signs near schools and other high-traffic areas. Despite the fact that a cohort of helmet-wearing riders had a higher probability of fatal crashes in 2018, it lowered the chance of fatal injury in 2020. As a result, a consistent campaign for helmet-wearing should be implemented, both in the form of public relations in schools and villages and through various online media.

Excessive speed continues to be a prevalent cause of accidents, posing a significant danger of fatalities. As a result, ongoing efforts to instill a sense of discipline in terms of traffic speed are essential to raising riders’ awareness of the severity of potential injuries. This can be achieved through multimedia campaigns that illustrate the correlation between speed and impact force. Agencies such as the Thai Health Promotion Foundation can leverage various media channels or websites to disseminate this information. In terms of traffic engineering, it is recommended to consider implementing road markings that promote lower speeds, such as those that say, for example, “Reduce speed in horizontal curves.” In areas with high traffic volumes, such as urban areas, traffic calming measures may be necessary [69]. Throughout all three years, traffic violations were identified as the primary cause of death, and the risk of fatalities was found to be increasing. One possible solution to address this behavior is to strengthen traffic discipline by identifying areas most prone to traffic signal violations and implementing corrective measures in those areas. Developing countries often face challenges such as excessive posters, which are often obstructed by billboards, and the degradation of various traffic warning signs such as those for curves, intersections, and speed limits. As a result, road safety engineers should undertake a thorough analysis and develop appropriate solutions. Illegal overtaking maneuvers and wrong-direction driving are both factors that contribute to a high likelihood of fatalities, but these incidents may only occur at specific risk points. Thus, a recommendation is to review previous crash data records and analyze the locations of regular crashes to identify potential solutions, such as adding solid lines, erecting signs prohibiting wrong-direction driving, etc.

The study found that crashes during nighttime hours, regardless of the presence of lighting, resulted in significantly higher crash severity due to the limited budget allocated to local roads. This results in inadequate lighting compared to larger roads prioritizing mobility. To address this issue, it is important to identify areas where crashes are likely to occur at night and to install appropriate lighting there. The local government organization is best suited to undertake this task and can collaborate with educational agencies or academic institutions to determine the most hazardous areas. In addition, relevant agencies should consider installing highly visible line markings and pavement markings on roads to enhance road safety.

The road characteristics variables revealed that roads undergoing construction or maintenance pose a high risk of death. Thus, it is recommended that public relations efforts be made to inform individuals in the area of the risks associated with such construction sites, and standardized signs should be placed in accordance with engineering principles. Moreover, a contracting agency should conduct regular inspections based on road safety audit principles. As for curved areas, the risk was only significant in 2018 and 2019. However, it is important that responsible organizations regularly assess the suitability of curves, including factors such as visibility, the speed when entering curves, and curve warning signs.

This study is a preliminary study of motorcycle crashes on local roads, which included those in both urban and rural areas. Therefore, there are some issues that require further study, such as the comparison of motorcycle crash severity in urban and rural areas. Determining an appropriate budget for road safety promotion requires considering economic aspects (i.e., where people are willing to pay for safer roads), as highlighted in previous studies [70]. For local roads, this study provides comprehensive recommendations on an overview level, but more precise analysis is required in each area to make more specific policy recommendations [71].

## Figures and Tables

**Figure 1 ijerph-20-03845-f001:**
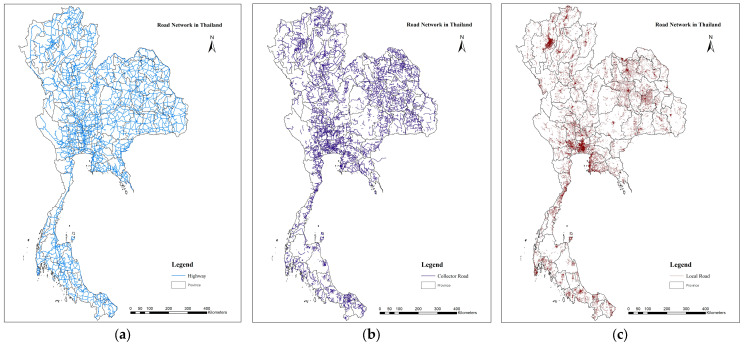
Road Network in Thailand. In 2021: (**a**) Length of Highway = 52,189 km (7.43%), (**b**) Collector Road = 49,124 km (6.99%), and (**c**) Local Road = 601,451 km (85.58%).

**Figure 2 ijerph-20-03845-f002:**
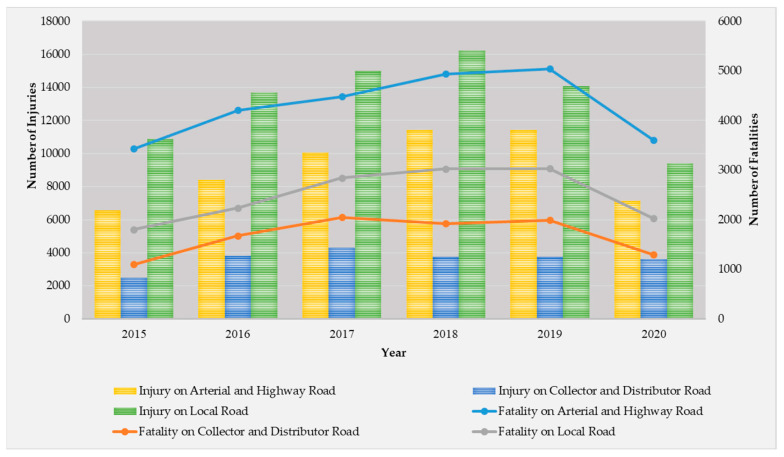
Number of Victims of Motorcycle Crashes among Road Types in Thailand in 2015–2020.

**Figure 3 ijerph-20-03845-f003:**
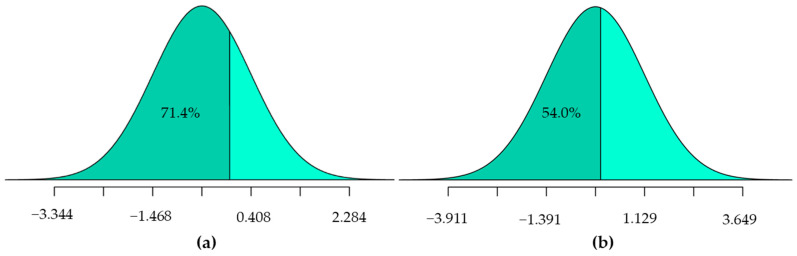
Normal distribution plot. (**a**) Foreign variables in 2019 model; (**b**) helmet variables in 2020 model.

**Table 1 ijerph-20-03845-t001:** Three Years of Previous Research: Analyses of Injury Severity for Motorcycle Crashes.

Study (Road Type)	Research Aims	Country	Methods
This study (Local roads)	This study focused on studying motorcycle crashes on local roads by using a sophisticated statistical model that included both unobserved heterogeneity and temporal instability analysis.	Thailand	Temporal instability random parameter logit model with unobserved heterogeneity in means and variances
Se, et al. [27] (Highway)	This study aimed to study the severity of motorcycle crashes, separating them into several models and classifying them by date of occurrence (weekday, weekend, and holiday) to illustrate the complexity of the model. The highlight is that time differences were also analyzed.	Thailand	Random-parameter logit model with unobserved heterogeneity in means and variances
Thongnak, et al. [28] (All types)	This study analyzed retrospective crash data using in-depth motorcycle crash data, extracted factors that affected the severity of the crashes, and summarized the results of the in-depth crash data with appropriate methods.	Thailand	Chi-square goodness of fit
Se, et al. [12] (Highway)	This study compared two analytical methods with injury outcomes for motorcycle crashes, including artificial neural networks and a random parameter logit model with unobserved heterogeneity in means and variances.	Thailand	Artificial neural networks and mixed logit model with unobserved heterogeneity in means and variances
Pervez, et al. [7] (All types)	This study focused on factors affecting fatalities in single-vehicle motorcycle crashes.	Pakistan	Random parameter model with heterogeneity in means and variances
Champahom, et al. [10] (Arterial road)	This study aimed to determine the factors affecting the level of injury in motorcycle crashes focusing on arterial roads only.	Thailand	Multiple correspondence analysis and ordered logistics regression
Islam [16] (Highway)	This study attempted to determine the factors influencing injuries in motorcycle crashes with a specific area as a work zone.	Florida, USA	Random parameter multinomial logit with heterogeneity in means and variances
Se, et al. [29] (Highway)	This study attempted to compare the severity of motorcycle injury levels. It was divided into two models: city road crashes and rural road crashes but not highway crashes.	Thailand	A correlated random parameter ordered probit approach with heterogeneity in means
Li, et al. [30] (Highway)	This study aimed to compare the severity levels of injuries from motorcycle crashes between one-, two-, and multi-car collisions.	California, USA	Latent class clustering and ordered probit model
Farid, et al. [25] (All types)	This study divided its model into two categories to determine the factors influencing fatalities from motorcycle crashes: single-vehicle crashes and multiple-vehicle crashes.	Wyoming, USA	Random parameters logistic regression with unobserved heterogeneity effects
Rezapour, et al. [26] (All types)	This study tried to find out the best way to figure out how bad a motorcycle crash is. It used several methods, including nonparametric machine leaning and parametric models.	Wyoming, USA	Random forest regression, support vector machine, and logistic regression mode
Rezapour, et al. [31] (All types)	This study attempted to determine the factors affecting the degree of injury in motorcycle crashes by using a variety of relationship-finding techniques both in machine learning and statistical modeling with parameter estimates.	Wyoming, USA	Decision tree and logistic regression
Abrari Vajari, et al. [32] (All types)	This study focused on analyzing the injury levels of motorcycle crashes at intersections.	Victoria, Australia	Multinomial logit model

**Table 2 ijerph-20-03845-t002:** A review of the findings in previous studies on the effects of various factors on motorcycle crash injury severities.

Variables	Impact on Motorcyclist Injury Severity
Rider characteristics	The rider’s individual traits make up a set of personal characteristics, and various factors can have varying impacts on the rider’s chances of mortality.
Gender (1 if male, 0 if female)	Male riders are more prone to engaging in hazardous driving practices, such as aggressive driving and speeding, as opposed to female riders [25,33,34,35,36,37,38].
Young (1 if rider was under the age of 30, 0 otherwise)	Driving behavior is influenced by age, and younger riders are physically stronger than middle-aged and elderly riders [6,25,31,33,37].
Old (1 if rider was over the age of 50, 0 otherwise)	Elderly riders generally exhibit slower reaction times, decreased decision-making abilities, and lower physical strength compared to middle-aged and young riders [6,25,31,39].
Local address (1 if rider’s address is the same district as the crash scene, 0 otherwise)	Riders who are familiar with the district where a crash occurs are aware of potential dangers and the likelihood of frequent accidents, which distinguishes them from riders who are unfamiliar with the area [32].
Foreign (1 if rider is foreign, 0 otherwise)	Foreign riders with a Thai driving license are allowed to drive, but they may not have the same level of familiarity with traffic regulations as Thai citizens [33].
Helmet (1 if rider uses a helmet, 0 otherwise)	Helmets are instruments that have the potential to lower the probability of fatality and mitigate the severity of injuries [15,32,37,40,41,42].
License (1 if rider has a rider’s license, 0 otherwise)	Obtaining a rider’s license entails being knowledgeable about and adhering to traffic regulations, resulting in a reduced likelihood of violating them [42,43].
Maneuvers prior to the crash	These variables related to the collision’s characteristics are predominantly obtained from police reports, as their purpose is to identify the cause of the accident.
Exceeding speed limit (1 if the cause of the crash was exceeding the speed limit, 0 otherwise)	Operating a vehicle at a speed that exceeds the limit can result in an increased risk of fatality, as the impact force generated by collisions or falls can be substantial [37].
Disobeying traffic sign (1 if the cause of the crash was disobeying a traffic sign, 0 otherwise)	Disregarding traffic signs and signals can indicate the location of a crash, such as at an intersection, and may contribute to the heightened or lowered probability of fatality [6,44].
Wrong direction (1 if the cause of the crash was riding in the wrong direction, 0 otherwise)	Erroneous driving in the opposite direction increases the risk of fatality in motorcycle accidents, which is primarily because such incidents often involve head-on collisions with properly driven cars [44].
Illegal overtaking (1 if the cause of the crash was an illegal overtaking maneuver, 0 otherwise)	Performing an illegal overtaking maneuver can elevate the likelihood of fatality, as it often requires acceleration to achieve high speeds during the maneuver [29].
Mobile use (1 if the cause of the crash was use of a mobile phone, 0 otherwise)	Motorcycle riders typically do not use their phones while driving; however, if an accident does occur due to phone use, the vehicle’s speed is likely to be lower. Consequently, there may be a chance of only minor injuries [34].
Asleep (1 if the cause of the crash was microsleep, 0 otherwise)	If a motorcycle rider experiences microsleep and collides with a different type of vehicle, then the likelihood of fatality is heightened [6,45].
Suddenly cutting (1 if the cause of the crash was suddenly cutting across lanes, 0 otherwise)	The impact of abrupt cutting off on the probability of fatality is ambiguous because its effect can be either increased or decreased depending on the direction and speed of the vehicles involved in the collision [44].
Drunk rider (1 if the cause of the crash was drunk driving, 0 otherwise)	A motorcyclist who is intoxicated has the potential to either increase or decrease the probability of fatality. Although some inebriated riders may simply fall off of their motorcycles, a collision with a different type of vehicle can result in a high likelihood of fatality [35,36,41,43].
Temporal and Environmental Characteristics	The timing of a crash can have an uneven effect on traffic volume, causing the probability of fatality to vary between motorcycle accidents. Additionally, driving visibility can also influence the severity of the rider’s injuries.
Festival duration (1 if the crash occurred during a festival period, 0 otherwise)	During the Songkran and New Year festivities, a prolonged period of celebration, most casualties in motorcycle accidents are typically drunk riders [29].
Morning peak hour (1 if the crash occurred between 7:00 a.m. and 8:00 a.m., 0 otherwise)	A crash that takes place during the morning rush hour may result in only a minor collision because the speed of the vehicles is usually not very high [25,32,46].
Afternoon peak hour (1 if the crash occurred between 4:00 p.m. and 5:00 p.m., 0 otherwise)	Motorcycle crashes that occur during the evening rush hour may be more hectic compared to those during the morning peak period [25,32,46].
Daytime (1 if the time of the crash was during the day, 0 otherwise)	Motorcycle accidents that happen during the day often involve high speeds, which consequently increases the probability of fatality [30,42,46,47].
Nighttime with light (1 if the time of crash was at night with lighting conditions, 0 otherwise)	Driving speed during the night can be influenced by visibility, and this may vary depending on the driving habits of each individual [37,46].
Darkness (1 if the crash was on a dark road, 0 otherwise)	In the event of a motorcycle accident at night without proper lighting, there is a possibility that the rider may not be able to brake the vehicle in time due to limited visibility [29,37,46,48].
Smoke (1 if the crash was in smoke, dust, or fog, 0 otherwise)	Smoke or dust can impair driving visibility, which can in turn affect driving speed.
Raining (1 if, at the time of the crash, it was raining, 0 otherwise)	Rain can impact driving speed as it increases the likelihood of both reduced visibility and slippery roads [29,32].
Road characteristics	The condition of the road surface and the physical characteristics of the roadway can have an impact on driving speed.
Under maintenance (1 if the surface of the road was under maintenance, 0 otherwise)	Accidents occurring in road maintenance areas, where traffic speed is typically reduced, may involve fixed objects at certain points [16].
Rough surface (1 if the surface of the road was rough, 0 otherwise)	Roads with bumps or potholes can have an impact on driving speed [41,43,49].
Curve (1 if the road was curved, 0 otherwise)	Motorcycle accidents that occur on curves are more likely to heighten the risk of fatality than those that take place on straight paths [6,30,38,41,46,50].
Dry (1 if the surface of the road was dry, 0 otherwise)	Riders may feel more confident driving on dry roads, and they tend to drive faster on them compared to wet roads [6,37,46].
Wet (1 if the surface of the road was wet, 0 otherwise)	Riders tend to exercise more caution and drive at lower speeds on wet roads [25,47].

**Table 3 ijerph-20-03845-t003:** Descriptive Statistics.

Variable	2018		2019		2020	
	Mean	S.D.	Mean	S.D.	Mean	S.D.
Fatal Crash (Fatal/Non-fatal crash)	2706/14,066		2719/12,398		1863/8313	
Rider characteristics						
Gender (1 if male, 0 if female)	0.742	0.437	0.748	0.434	0.754	0.430
Young (1 if the rider was under the age of 30, 0 otherwise)	0.366	0.482	0.338	0.473	0.327	0.469
Old (1 if the rider was over the age of 50, 0 otherwise)	0.253	0.435	0.302	0.459	0.299	0.458
Local address (1 if the rider’s address is the same district as the crash scene, 0 otherwise)	0.747	0.435	0.700	0.458	0.748	0.434
Foreign (1 if the rider was foreign, 0 otherwise)	0.025	0.155	0.023	0.149	0.027	0.162
Helmet (1 if the rider used a helmet, 0 otherwise)	0.517	0.500	0.519	0.500	0.375	0.484
License (1 if rider has a rider’s license, 0 otherwise)	0.969	0.174	0.969	0.173	0.964	0.186
Maneuvers prior to the crash						
Exceeding speed limit (1 if the cause of the crash was exceeding the speed limit, 0 otherwise)	0.286	0.452	0.318	0.466	0.250	0.433
Disobeying traffic sign (1 if the cause of the crash was disobeying a traffic sign, 0 otherwise)	0.005	0.074	0.008	0.089	0.006	0.078
Wrong direction (1 if the cause of the crash was riding in the wrong direction, 0 otherwise)	0.003	0.057	0.003	0.056	0.003	0.052
Illegal overtaking (1 if the cause of the crash was an illegal overtaking maneuver, 0 otherwise)	0.011	0.105	0.006	0.077	0.005	0.069
Mobile use (1 if the cause of the crash was using a mobile phone, 0 otherwise)	0.001	0.035	0.011	0.104	0.002	0.046
Asleep (1 if the cause of the crash was microsleep, 0 otherwise)	0.004	0.064	0.005	0.069	0.006	0.076
Suddenly cutting (1 if the cause of the crash was suddenly cutting lanes, 0 otherwise)	0.148	0.355	0.197	0.398	0.114	0.318
Drunk rider (1 if the cause of the crash was drunk riding, 0 otherwise)	0.181	0.385	0.151	0.358	0.176	0.381
Temporal and Environmental Characteristics						
Festival duration (1 if the crash occurred during a festival period, 0 otherwise)	0.185	0.388	0.180	0.384	0.173	0.378
Morning peak hour (1 if the crash occurred between 7:00 a.m. and 8:00 a.m., 0 otherwise)	0.076	0.266	0.068	0.253	0.079	0.270
Afternoon peak hour (1 if the crash occurred between 4:00 p.m. and 5:00 p.m., 0 otherwise)	0.144	0.351	0.139	0.346	0.132	0.339
Daytime (1 if the time of the crash was during the day, 0 otherwise)	0.563	0.496	0.534	0.499	0.549	0.498
Nighttime with light (1 if the time of crash was at night with lighting conditions, 0 otherwise)	0.186	0.389	0.185	0.388	0.215	0.411
Darkness (1 if the crash was on a dark road, 0 otherwise)	0.199	0.399	0.170	0.376	0.190	0.392
Smoke (1 if the crash was in smoke, dust, or fog, 0 otherwise)	0.028	0.165	0.028	0.164	0.029	0.168
Raining (1 if, at the time of the crash, it was raining, 0 otherwise)	0.038	0.192	0.017	0.128	0.031	0.175
Road characteristics						
Under maintenance (1 if the surface of the road was under maintenance, 0 otherwise)	0.001	0.039	0.001	0.024	0.002	0.050
Rough surface (1 if the surface of the road was rough, 0 otherwise)	0.030	0.170	0.024	0.152	0.027	0.163
Curve (1 if the road was curved, 0 otherwise)	0.142	0.349	0.129	0.335	0.145	0.352
Dry (1 if the surface of the road was dry, 0 otherwise)	0.852	0.355	0.833	0.373	0.877	0.328
Wet (1 if the surface of the road was wet, 0 otherwise)	0.046	0.210	0.024	0.154	0.033	0.178

**Table 4 ijerph-20-03845-t004:** The results of likelihood ratio tests for different time periods.

*m* _1_	*m* _2_		
	2018	2019	2020
2018	−	43.94 (30)	54.97 (34)
		[95.16%]	[98.71%]
2019	43.33 (29)	−	169.143 (34)
	[95.76%]		[99.99%]
2020	48.67 (29)	72.49 (30)	−
	[98.75%]	[99.99%]	

**Table 5 ijerph-20-03845-t005:** Goodness of fit.

Model Statistics	Year			
	2018–2020	2018	2019	2020
Logit Model				
N	42,065	16,772	15,117	10,176
*LL*(β)	−17,221.65	−6406.92	−6256.17	−4243.65
*LL*(0)	−19,392.43	−7411.31	−7122.93	−4844.07
*k*	29	29	29	29
McFadden *ρ^2^*	0.112	0.136	0.122	0.124
Adjusted McFadden *ρ^2^*	0.110	0.132	0.118	0.118
Random parameter Logit Model with Unobserved Heterogeneity with Means and Variances				
*LL*(β)	−17,151.65	−6368.00	−6193.17	−4202.20
*LL*(0)	−19,392.43	−7411.30	−7122.93	−4844.06
*k*	31	29	30	34
McFadden *ρ^2^*	0.116	0.141	0.131	0.133
Adjusted McFadden *ρ^2^*	0.114	0.137	0.126	0.125

**Table 6 ijerph-20-03845-t006:** Results of motorcycle crash fatality analysis for crashes in 2018.

Variable	Parameter Estimation	t−Stat
Constant	−1.708	−16.79
Rider characteristics		
Gender (1 if male, 0 if female)	0.285	1.02
Standard deviation of “Festival duration” (Normal distribution)	0.076	2.74
Young (1 if the rider was under the age of 30, 0 otherwise)	−0.186	−4.61
Old (1 if rider was over the age of 50, 0 otherwise)	0.346	7.99
Local address (1 if the rider’s address was in the same district as the crash scene, 0 otherwise)	−0.729	−2.80
Standard deviation of “Local address” (Normal distribution)	0.234	2.55
Foreign (1 if the rider was foreign, 0 otherwise)	0.229	2.20
Helmet (1 if the rider used a helmet, 0 otherwise)	1.117	3.78
Maneuvers prior to the crash		
Exceeding speed limit (1 if the cause of the crash was exceeding the speed limit, 0 otherwise)	0.966	26.74
Disobeying traffic sign (1 if the cause of the crash was disobeying a traffic sign, 0 otherwise)	0.738	3.82
Wrong direction (1 if the cause of the crash was riding in the wrong direction, 0 otherwise)	0.581	2.39
Asleep (1 if the cause of the crash was microsleep, 0 otherwise)	0.734	3.22
Suddenly cutting (1 if the cause of the crash was suddenly cutting across lanes, 0 otherwise)	−0.136	−2.53
Temporal and Environmental Characteristics		
Festival duration (1 if the crash occurred during a festival period, 0 otherwise)	−0.443	−1.26
Standard deviation of “Festival duration” (Normal distribution)	0.234	2.55
Morning peak hour (1 if the crash occurred between 7:00 a.m. and 8:00 a.m., 0 otherwise)	0.111	1.65
Afternoon peak hour (1 if the crash occurred between 4:00 p.m. and 5:00 p.m., 0 otherwise)	−0.139	−2.16
Daytime (1 if the time of the crash was during the day, 0 otherwise)	−0.331	−3.49
Smoke (1 if the crash was in smoke, dust, or fog, 0 otherwise)	−0.912	−6.59
Road characteristics		
Rough surface (1 if the surface of the road was rough, 0 otherwise)	−0.841	−5.50
Curve (1 if the road was curved, 0 otherwise)	0.089	1.76
Wet (1 if the surface of the road was wet, 0 otherwise)	−0.526	−2.44
Heterogeneity in the mean of the random parameters		
Festival duration: License (1 if the rider had a rider’s license, 0 otherwise)	−0.528	−1.75
Gender: Drunk rider (1 if the cause of the crash was drunk riding, 0 otherwise)	−0.432	−3.92
Gender: License (1 if rider had a rider’s license, 0 otherwise)	0.729	2.74
Gender: Dry (1 if the surface of the road was dry, 0 otherwise)	−0.400	−4.12
Local address: Dry (1 if the surface of the road was dry, 0 otherwise)	0.424	4.49
Heterogeneity in the variance of the random parameters		
Local address: Afternoon peak hour (1 if the crash occurred between 4:00 p.m. and 5:00 p.m., 0 otherwise)	3.061	34.14
Model Statistics		
Log−likelihood at convergence (*β*)	−6368.001	
Log−likelihood at zero (0)	−7411.308	
Degrees of freedom	29	
McFadden *ρ*^2^	0.141	
Adjusted McFadden *ρ^2^*	0.137	

**Table 7 ijerph-20-03845-t007:** Results of motorcycle crash fatality analysis for crashes in 2019.

Variable	Parameter Estimation	t−Stat
Constant	−1.537	−22.05
Rider characteristics		
Gender (1 if male, 0 if female)	0.541	12.31
Young (1 if the rider was under the age of 30, 0 otherwise)	−0.137	−3.46
Local address (1 if the rider’s address was in the same district as the crash scene, 0 otherwise)	−0.062	−1.69
Foreign (1 if rider was foreign, 0 otherwise)	−0.530	−0.14
Standard deviation of “Foreign” (Normal distribution)	0.938	4.98
Maneuvers prior to the crash		
Mobile use (1 if the cause of the crash was using a mobile phone, 0 otherwise)	−2.487	−3.51
Suddenly cutting (1 if the cause of the crash was suddenly cutting across lanes, 0 otherwise)	−0.477	−9.63
Temporal and Environmental Characteristics		
Festival duration (1 if the crash occurred during the festival period, 0 otherwise)	−0.561	−1.22
Standard deviation of “Festival” (Normal distribution)	0.229	2.38
Morning peak hour (1 if the crash occurred between 7:00 a.m. and 8:00 a.m., 0 otherwise)	0.154	2.36
Afternoon peak hour (1 if the crash occurred between 4:00 p.m. and 5:00 p.m., 0 otherwise)	−0.093	−1.66
Daytime (1 if the time of the crash was during the day, 0 otherwise)		
Nighttime with light (1 if the time of crash was at night with lighting conditions, 0 otherwise)	0.736	8.14
Darkness (1 if the time of the crash was on a dark road, 0 otherwise)	0.771	8.53
Road characteristics		
Rough surface (1 if the surface of the road was rough, 0 otherwise)	−0.439	−3.36
Dry (1 if the surface of the road was dry, 0 otherwise)	−1.854	−5.40
Standard deviation of “Dry” (Normal distribution)	0.326	10.59
Heterogeneity in the mean of the random parameters		
Festival duration: Wet (1 if the surface of the road was wet, 0 otherwise)	−0.993	−2.69
Festival duration: Helmet (1 if the rider used a helmet, 0 otherwise)	0.285	2.26
Festival duration: Daytime (1 if the time of the crash was during the day, 0 otherwise)	0.285	2.36
Dry: Drunk rider (1 if the cause of the crash was drunk riding, 0 otherwise)	−0.618	−8.06
Dry: Helmet (1 if the rider used a helmet, 0 otherwise)	−0.229	−4.76
Dry: License (1 if the rider has a riding license, 0 otherwise)	1.537	4.64
Dry: Daytime (1 if the time of the crash was during the day, 0 otherwise)	0.242	2.58
Dry: Old (1 if the rider was over the age of 50, 0 otherwise)	0.318	6.21
Foreign: Drunk rider (1 if the cause of the crash was drunk riding, 0 otherwise)	−0.741	−2.32
Heterogeneity in the variance of the random parameters		
Dry: Young (1 if the rider was under the age of 30, 0 otherwise)	−0.748	−4.34
Model Statistics		
Log−likelihood at convergence (*β*)	−6193.171	
Log−likelihood at zero (0)	−7122.931	
Degrees of freedom	30	
McFadden *ρ*^2^	0.131	
Adjusted McFadden *ρ*^2^	0.126	

**Table 8 ijerph-20-03845-t008:** Results of motorcycle crash fatality analysis for crashes in 2020.

Variable	Parameter Estimation	t−Stat
Constant	−2.108	−17.59
Rider characteristics		
Gender (1 if male, 0 if female)	0.191	0.70
Standard deviation of “Gender” (Normal distribution)	0.163	2.61
Young (1 if the rider was under the age of 30, 0 otherwise)	−0.299	−5.76
Old (1 if the rider was over the age of 50, 0 otherwise)	0.142	2.80
Local address (1 if the rider’s address was in the same district as the crash scene, 0 otherwise)	−0.942	−3.26
Foreign (1 if the rider was foreign, 0 otherwise)	0.290	2.44
Helmet (1 if the rider used a helmet, 0 otherwise)	−1.131	−1.63
Standard deviation of “Helmet” (Normal distribution)	1.260	13.05
Maneuvers prior to the crash		
Exceeding speed limit (1 if the cause of the crash was exceeding the speed limit, 0 otherwise)	0.891	19.96
Disobeying traffic sign (1 if the cause of the crash was disobeying a traffic sign, 0 otherwise)	0.435	1.79
Wrong direction (1 if the cause of the crash was riding in the wrong direction, 0 otherwise)		
Illegal overtaking (1 if the cause of the crash was an illegal overtaking maneuver, 0 otherwise)	1.488	5.71
Suddenly cutting (1 if the cause of the crash was suddenly cutting across lanes, 0 otherwise)	−0.194	−2.73
Drunk rider (1 if the cause of the crash was drunk riding, 0 otherwise)	−0.975	−1.93
Standard deviation of “Drunk” (Normal distribution)	0.428	2.46
Temporal and Environmental Characteristics		
Festival duration (1 if the crash occurred during the festival period, 0 otherwise)	−0.092	−0.22
Standard deviation of “Festival” (Normal distribution)	1.176	5.79
Nighttime with light (1 if the time of crash was at night with lighting conditions, 0 otherwise)	0.299	2.98
Road characteristics		
Under maintenance (1 if the surface of the road was under maintenance, 0 otherwise)	1.555	3.97
Rough surface (1 if the surface of the road was rough, 0 otherwise)	0.798	4.74
Dry (1 if the surface of the road was dry, 0 otherwise)	0.455	4.26
Heterogeneity in the mean of the random parameters		
Festival duration: License (1 if rider had a rider’s license, 0 otherwise)	−1.380	−3.46
Festival duration: Smoke (1 if the crash was in smoke, dust, or fog, 0 otherwise)	0.592	1.66
Drunk rider: Daytime (1 if the time of the crash was during the day, 0 otherwise)	0.401	2.71
Drunk rider: Wet (1 if the surface of the road was wet, 0 otherwise)	1.546	2.28
Drunk rider: Raining (1 if, at the time of the crash, it was raining, 0 otherwise)	−1.358	−1.88
Helmet: Mobile use (1 if the cause of the crash was using a mobile phone, 0 otherwise)	−2.912	−2.57
Helmet: Curve (1 if the road was curved, 0 otherwise)	0.248	2.00
Helmet: Raining (1 if, at the time of the crash, it was raining, 0 otherwise)	−0.887	−2.12
Helmet: Smoke (1 if the crash was in smoke, dust, or fog, 0 otherwise)	−0.655	−1.83
Gender: License (1 if rider had a rider’s license, 0 otherwise)	0.583	2.21
Heterogeneity in the variance of the random parameters		
Helmet: Local address (1 if the rider’s address was in the same district as the crash scene, 0 otherwise)	0.698	7.92
Helmet: Afternoon peak hour (1 if the crash occurred between 4:00 p.m. and 5:00 p.m., 0 otherwise)	−6.263	−35.52
Helmet: Darkness (1 if the crash was on a dark road, 0 otherwise)	0.407	3.87
Model Statistics		
Log−likelihood at convergence (*β*)	−4202.207	
Log−likelihood at zero (0)	−4844.067	
Degrees of freedom	34	
McFadden *ρ*^2^	0.133	
Adjusted McFadden *ρ*^2^	0.125	

**Table 9 ijerph-20-03845-t009:** Temporal comparison of marginal effects.

Variable	Marginal Effect		
	2018	2019	2020
Rider Characteristics			
Gender (1 if male, 0 if female)	0.0809	0.0851	0.1143
Young (1 if the rider was under the age of 30, 0 otherwise)	−0.0291	−0.0293	−0.0517
Old (1 if the rider was over the age of 50, 0 otherwise)	0.0554		0.0633
Local address (1 if the rider’s address was in the same district as the crash scene, 0 otherwise)	−0.0116	−0.0139	0.0038
Foreign (1 if the rider was foreign, 0 otherwise)	0.0479		0.0685
Helmet (1 if the rider used a helmet, 0 otherwise)	0.0154		−0.0756
License (1 if the rider had a rider’s license, 0 otherwise)			
Maneuvers prior to the crash			
Exceeding speed limit (1 if the cause of the crash was exceeding the speed limit, 0 otherwise)	0.1744		0.2509
Disobeying traffic sign (1 if the cause of the crash was disobeying traffic signs, 0 otherwise)	0.1537	0.0157	0.0595
Wrong direction (1 if the cause of the crash was riding in the wrong direction, 0 otherwise)	0.1036		
Illegal overtaking (1 if the cause of the crash was an illegal overtaking maneuver, 0 otherwise)			0.4950
Mobile use (1 if the cause of the crash was using a mobile phone, 0 otherwise)		−0.1745	
Asleep (1 if the cause of the crash was microsleep, 0 otherwise)	0.1535		
Suddenly cutting (1 if the cause of the crash was suddenly cutting across lanes, 0 otherwise)	−0.0227	−0.0817	−0.0727
Drunk rider (1 if the cause of the crash was drunk riding, 0 otherwise)			−0.1186
Temporal and Environmental Characteristics			
Festival duration (1 if the crash occurred during the festival period, 0 otherwise)	−0.1226	−0.1368	−0.1937
Morning peak hour (1 if the crash occurred between 7:00 a.m. and 8:00 a.m., 0 otherwise)	0.0230	0.0445	
Afternoon peak hour (1 if the crash occurred between 4:00 p.m. and 5:00 p.m., 0 otherwise)	−0.0213	−0.0103	−0.0513
Daytime (1 if the time of the crash was during the day, 0 otherwise)	−0.0595		
Nighttime with light (1 if the time of crash was at night with lighting condition, 0 otherwise)		0.0853	0.0528
Darkness (1 if the crash was on a dark road, 0 otherwise)	0.0017	0.0914	0.0324
Smoke (1 if the crash was in smoke, dust, or fog, 0 otherwise)	−0.1025	−0.0219	
Raining (1 if, at the time of the crash, it was raining, 0 otherwise)		−0.0005	
Road characteristics			
Under maintenance (1 if the surface of the road was under maintenance, 0 otherwise)			0.3518
Rough surface (1 if the surface of the road was rough, 0 otherwise)	−0.0605	−0.0143	0.0822
Curve (1 if the road curved, 0 otherwise)	0.0046	0.0116	
Dry (1 if the surface of the road was dry, 0 otherwise)		0.005	0.1246
Wet (1 if the surface of the road was wet, 0 otherwise)	−0.0393		

## Data Availability

Data available on request due to restrictions.

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
