# Peer review of "Temporal Instability of Motorcycle Crash Fatalities on Local Roadways: A Random Parameters Approach with Heterogeneity in Means and Variances"

_ijerph, 2023, doi:10.3390/ijerph20053845_

Round 1

Reviewer 1 Report

This study explores the temporal instability of the factor coefficients in modeling motorcycle crashes. This issue has been widely analyzed recently, but the authors provide some new views on analyzing the local road crashes. However, this paper is still needed to be polished before the publication. Please see my comments.

1 As shown in Table 1 and 2, many pieces of evidence have analyzed the injury-level of motorcycle crashes. The inadequacy of these study is that they failed to consider the crashes occur on local roads. Thus, the difference of the characteristics between local roads and other roads/highways, as well as the corresponding crashes, are suggested to be described in the Introduction part.

2 To strengthen the literature review, some recent published papers that use the heterogeneity in means and variances approach are recommended to be discussed and mentioned such as

https://doi.org/10.1061/JTEPBS.0000717

https://doi.org/10.1016/j.amar.2020.100126

https://doi.org/10.1016/j.amar.2022.100244

https://doi.org/10.1016/j.amar.2021.100161

2 The summary of the key findings are recommended in the Introduction part or the Conclusions and Implementations part.

3 It seems that some factors listed in Table 3 are not properly categorized or named. For example, the time of day merely contains the “Morning peak hour” and “Afternoon peak hours”, while other periods such as “Daytime” and “Night” are considered as environment factors. The factor of “crash characteristics” could better be named as “maneuvers prior to the crash”.

4 It is strange that while exceeding the speed limit is significant in 2018 modeling with a very high t-value (26.74), this factor is not significant in 2019 and 2020 modeling. This is due to the coefficient mean is related to other factors in 2019, but why it is not significant in 2020?

5 The examination of multicollinearity and the Pearson correlation of the independent variables is recommended before the modeling.

6 Also, many studies have highlighted the importance of speeding on triggering fatal crashes and the necessity of considering speeding or vehicle speed on road safety analysis. Thus, the discussion of the relationship between vehicle speeding and crash fatality could be enhanced.

7 The English writing is supposed to be polished.

Reviewer 2 Report

roads. However, at current state the manuscript requires some improvements for its better quality:

- after well prepared literature review it would be worth to clearly present the research area, currently it’s not clear whether accident statistics consider all local roads or only rural/urban roads, this is all the more essential as the authors provide specific recommendations in their conclusions,

-          analyses period and conclusions drawn on this basis can be misleading due to specific/pandemic year 2020 (reduction of traffic led to reduction of accidents/fatalities all over the world),

-          table 3 is wrongly referenced,

-          wrong year (2022) is given in line 257,

-          some of authors’ statement may rise doubts, such as that given in chapter 7.2 where crashes occurring in the morning positively significant were explained by rush hours while afternoon crashes – negatively significant – were justified by traffic jams occurring in afternoon rush hour; it seems to be highly inconsistent. Additionally authors wrote (lines 480-481) that ME became increasingly negative over the time while the tendency is rather fluctuate.

-          Similarly dubious explanation is given in chapter 7.5 when crashes are explained by the road rough surface: in years 2018-2019 it has a positive influence and in 2020 negative influence.

-          Some of the recommendations given in chapter 8 are unconvincing, for instance the one given for morning peak hours – a publicity sign (line 616) or those for excessive speed and environmental variables – authors suggest some information campaigns whose effectiveness is questionable and do not suggest any solutions related to speed measurements like traffic calming or at least consideration of road markings which can be very helpful not only on secondary roads in poor, restricted or night visibility.

Round 2

Reviewer 1 Report

All the reviewer comments have been carefully addressed, and it is recommended for publication.